# The Writing Process and the Written Product in Bimodal Bilingual Deaf and Hard of Hearing Children

Moa Gärdenfors 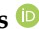

Department of Linguistics, Stockholm University, 10691 Stockholm, Sweden; moa.gardenfors@ling.su.se

**Abstract:** How does bimodal bilingualism—a signed and a spoken language—influence the writing process or the written product? The writing outcomes of twenty deaf and hard of hearing (DHH) children and hearing children of deaf adults (CODA) (mean 11.6 years) with similar bimodal bilingual backgrounds were analyzed. During the writing of a narrative text, a keylogging tool was used that generated detailed information about the participants' writing process and written product. Unlike earlier studies that have repeatedly shown that monolingual hearing children outperform their DHH peers in writing, there were few differences between the groups that likely were caused by their various hearing backgrounds, such as in their lexical density. Signing knowledge was negatively correlated with writing flow and pauses before words, and positively correlated with deleted characters, but these did not affect the written product negatively. Instead, they used different processes to reach similar texts. This study emphasizes the importance of including and comparing participants with similar language experience backgrounds. It may be deceptive to compare bilingual DHH children with hearing children with other language backgrounds, risking showing language differences. This should always be controlled for through including true control groups with similar language experience as the examined groups.

**Keywords:** deaf and hard of hearing; DHH; CODA; bimodal bilingualism; bilingualism; sign language; written product; writing process; keystroke logging; literacy

## 1. Introduction

Bimodal bilingualism, being bilingual in two different modalities, is characteristic of those who use a spoken language and a sign language. Hearing children of deaf adults (henceforth CODA) typically belong to this group because their deaf parents use sign language, while there is only a minority of deaf and hard of hearing children (henceforth DHH children) who speak and comprehend both a spoken and a signed language. The majority of the DHH children are born into hearing families who have never been in contact with sign language before, with the result that many of these children do not learn any sign language (Humphries et al. 2012; Mauldin 2012). An understudied area linked to this is the relationship of bimodal bilingualism and the DHH children's writing. The motivation of the study is therefore to explore the influence of the knowledge of both spoken Swedish and Swedish Sign Language on the written product and process of DHH and hearing bimodal bilingual children.

This is of interest because some previous studies have investigated literacy development in DHH sign–print bilingual children who use both a sign language and a written language but not a spoken language (see Humphries et al. 2013 for an overview). The findings from these studies suggest that early and consistent development of sign language facilitates literacy development for this group. However, the sign–print bilingual children differ from DHH bimodal bilingual children who speak and comprehend both sign language and spoken language, often thanks to their access to hearing technology. Still, the facilitating effect of sign language, found for the sign–print bilingual DHH group, has constituted a starting point for the present study. This study will therefore only include bimodal bilingual DHH participants with access to a spoken and a signed language.

Unlike 15–20 years ago, many DHH children nowadays are provided with hearing technologies such as hearing aids or cochlear implants (CI), and they have therefore received opportunities to develop and comprehend spoken language in a similar way as hearing children (SOU 2016, p. 46). As a consequence, this has resulted in a shifted educational and research interest in how well this DHH group performs in their spoken language, and how this performance is related to their literacy development. The result is a decreased interest in researching early sign language acquisition and bimodal bilingualism for this DHH group with hearing technology.

Another issue related to this knowledge gap is that several DHH studies have not taken their participants' bimodal bilingual backgrounds into consideration, but instead this group has often been compared to monolingual hearing controls, thus, not taking into account potential differences regarding their different linguistic backgrounds (e.g., Hall and Dills 2020). Furthermore, although there is a handful of studies about the bimodal bilingual effect on the literacy outcomes (notably Hassanzadeh 2012; Davidson et al. 2014; Amraei et al. 2017; Gärdenfors et al. 2019; Gärdenfors n.d.), there are almost no studies on how bimodal bilingualism affects the real time process of writing (such as writing speed, pause behavior and revisions; however, see Gärdenfors et al. 2019).

Why is a focus on the analysis of the real time writing process important? An analysis of a written text usually consists of a final written product only (that is, the written text), a study of the writing process can additionally give us information on what was easy or challenging for the writer during the text production: How long did it take for the writer to write it? Did the text undergo many revisions? Was the writer a confident speller? What did the writer's pause behavior look like? Two written products (i.e., two texts) can be very similar, regarding for instance text length and lexical variation, but the writing processes behind them can be completely different.

There are some indications that unimodal hearing bilinguals (of two spoken languages) and L2 writers show a slower writing behavior that uses more pauses, due to a higher cognitive load (Schoonen et al. 2009; Palviainen et al. 2012); it is of interest to study if this also applies to bimodal bilinguals using a spoken and a signed language.

The aim of this study is to explore if, and how, bimodal bilingualism affects the writing outcomes among DHH and CODA children. In particular, the study aims to compare the written products and the writing processes between DHH and CODA groups, both using spoken Swedish and Swedish Sign Language (*Svenskt teckenspråk*, henceforth, STS). This study will, on the one hand, compare the children's writing processes and, on the other hand, their written products. For this purpose, ten DHH children (mean age 11.7) are compared to ten hearing CODAs (mean age 11.4); both groups are bimodal bilingual in spoken Swedish and STS.

## 2. Bimodal Bilingualism

This section introduces the concept of bimodal bilingualism and how it is related to other kinds of bilingualism. Bimodal bilingualism, which is central to this study, has similarities and differences compared to unimodal bilingualism. The languages in a bimodal bilingualism come from two different modalities—such as from a sign language and a spoken language. In contrast, the languages in a unimodal bilingualism derive from one and the same modality—such as two spoken languages or two sign languages.

Despite the modality difference, bimodal bilingual speakers and unimodal bilinguals share a number of similarities. A neuropsychological finding is that highly proficient bimodal and unimodal bilinguals constantly have to suppress one language while using the other language, because both languages are constantly active (Morford et al. 2011; Chabal and Marian 2015). This constant suppression of the "unwanted" language has been suggested as a disadvantage in their lexical retrieval (i.e., how fast they find a word) compared to their monolingual peers (Bialystok et al. 2008; Bialystok 2009; Duñabeitia et al. 2013; Giezen and Emmorey 2017). From a L2 view, this suppression has also been mirrored in writing, showing that L2 writers spend twice as much time on text composing compared

to L1 writers, which has been explained by slower retrieval time for L2 writers (de Lario et al. 2006; Lindgren et al. 2019). Moreover, both the unimodal and the bimodal bilinguals have access to two cultural and linguistic repertoires, and will therefore have double access to metalinguistic knowledge in two languages, while the monolinguals only have access to one (Williams and Lowrance-Faulhaber 2018). There is at least one crucial circumstance that differentiates bimodal bilinguals from unimodal bilinguals: the bimodal bilinguals of a spoken and a signed language are able to express both languages simultaneously, also known as code-blending (expressing signs and spoken words at the same time), while it is physically impossible for the unimodal bilinguals to express two spoken languages simultaneously, so instead they must code-switch (stop using one language to start using the another). Even if bimodal bilinguals can both code-blend and code-switch, studies have found that they prefer code-blending to code-switching; code-blending has the advantage of enabling them to express both representations simultaneously, while code-switching requires them to suppress the other language (Emmorey et al. 2008b).

The research regarding bimodal bilingualism has primarily been based on CODAs rather than the DHH group (Emmorey et al. 2008a, 2008b; Giezen et al. 2015; Giezen and Emmorey 2017). This is because there are few DHH children mastering both a spoken and a signed language. Their lack of sign language knowledge can often be attributed to the fact that the children's hearing parents are not often encouraged to learn, or to teach their children, sign language. Instead, the children's hearing parents are often concerned that the training of sign language occurs at the expense of speech training (Mauldin 2012; Humphries et al. 2012). As mentioned in the introduction section, another issue is that many studies do not differentiate among DHH children depending on their degree of competence in a sign language, but instead collapsing all DHH children with varying linguistic backgrounds into one group (Hall and Dills 2020).

### 3. Bimodal Bilingual Studies of the DHH and CODA Group

This section covers the current situation of the DHH group as well as bimodal bilingual studies of DHH and CODA groups.

Since new hearing technologies were widely introduced 15–20 years ago, more than 95% of the DHH children in Sweden are now provided with cochlear implants (CI) or digital hearing aids, enabling them to develop and comprehend spoken language (SOU 2016, p. 46). This has, in turn, resulted in fewer bimodal bilingual DHH children, which explains why there are so few studies regarding the impact of sign language knowledge in the DHH group.

The few bimodal bilingual-based DHH studies to date have, however, shown that learning a sign language does not inhibit simultaneously learning a spoken language (Hassanzadeh 2012; Amraei et al. 2017). This is in line with studies on sign–print bilingual DHH children showing that an early and consistent access to sign language facilitated their literacy development, and that they consistently outperformed their DHH peers who never learned sign language or started learning sign language later in life (Strong and Prinz 1997; Svartholm 2006; Mayberry 2007; Humphries et al. 2013).

One study (Davidson et al. 2014) compared the speech and language outcomes in five 4–7-year-olds American DHH children using CI with twenty age-matched CODAs, both groups understanding spoken English and American sign language (ASL). Both groups performed equally on all language measures and the authors concluded that sign language does not impair the auditory or language development of the DHH group (Davidson et al. 2014).

Goodwin et al. (2017) examined the use of the articles *a/n* and *the* in spoken English in 3–5-year-old bimodal bilingual children, who had been exposed to ASL from birth, who neither were CODA nor DHH using CI. The rationale for the study was that since ASL lacks articles, the authors wanted to examine if they would find any transfer from ASL, which would result in a lower incidence of articles in spoken English. This study also included two control groups of hearing monolinguals and unimodal bilinguals of Cantonese and English, which is an interesting comparison because Cantonese also lacks articles. The result showed

that the DHH group did not differ from the CODA or the hearing monolinguals while the Cantonese–English group omitted more obligatory articles. The authors concluded that sign language knowledge, even if it lacks some linguistic patterns, does not impair the development of spoken language among the bimodal bilingual group.

With regard to the effect of sign language on the bimodal bilingualism of the DHH groups, Gärdenfors (n.d.) examined the writing development, with a particular focus on the ability to describe words. The participants included sixteen DHH students in the age span 8–18 using hearing aids or CI. Half of them were bimodal bilinguals who were age-matched to the other half, who were monolingual in spoken language. The study found that the bimodal bilingual DHH students outperformed the monolingual DHH students on all text measures, and that they used significantly more descriptive words (such as adjectives and adverbs) as well as demonstrating a higher lexical diversity (i.e., number of unique words) compared to the monolingual DHH group. This is interesting because much of the descriptive information is expressed non-manually and facially through facial expressions and is not always included in the signs. The study proposed that the bimodal bilinguals' early access to sign language may have contributed to the advantage in literacy development, as well as their access to a bilingual and a bicultural repertoire.

Another study examined spelling in 33 Swedish children with varying backgrounds in hearing and language. The participating children were either sign–print bilingual DHH, bimodal bilingual DHH children, CODA, unimodal bilinguals or hearing monolinguals. The authors found that the DHH and CODA groups performed similarly in their spelling errors, which were both auditory and visually based. The visually based errors could neither be found among the unimodal bilinguals nor the hearing monolinguals. This was explained as a result of the children's early exposure to fingerspelling from their signing or DHH parents. The parents would typically explain words through fingerspelling. This habit provided the children with an additional strategy to circumvent the sounding strategy of spelling. The same study also suggested that there is a transfer from sign language and fingerspelling to writing: the sign–print bilingual DHH children were found to use some letters incorrectly in a word, because they were misled by the corresponding sign that happened to have this letter as handshape. This phenomenon was, however, not found among the bimodal bilingual DHH and CODA group using spoken and sign language (Gärdenfors et al. 2019).

Typical transfer from sign language among the DHH group has been found in a range of other studies where it does not only concern spelling (see also Sutcliffe et al. 1999) but also grammar and morphosyntax (e.g., Wolbers et al. 2014a, 2014b). Some literature has indicated that CODAs' language development is similar to the language acquisition of children with a migration background (Hofmann and Chilla 2015), while the majority of the studies argues that CODAs follow the same developmental pattern in their second language as their monolingual peers (see, e.g., Singleton and Tittle 2000; Brackenbury et al. 2006).

## 4. Written Product

This section covers written product outcomes reported by earlier studies on the DHH groups with varying language backgrounds, with a particular focus on number of words, lexical density, lexical diversity and spelling errors.

*Number of words.* The DHH group has repeatedly been reported to write fewer words, in the form of shorter texts compared to their monolingual and bilingual hearing peers. This is thought to be caused by a smaller vocabulary and a limited auditory input (e.g., Arfé et al. 2016; Gärdenfors et al. 2019; Oliveira et al. 2020).

*Lexical density* is a way to examine the proportion of content words (lexical words such as nouns, verbs, adjectives and some lexical adverbs), and function words (grammatical words such as pronouns, prepositions, interjections, conjunctions and numerals) in a text. A high lexical density shows that a text has many lexical words, which is a property that is more associated with written language. Vice versa, a low lexical density indicates that a text has many fillers and grammatical words that are more associated with spoken language

(Halliday 1985; Wengelin 2006; Johansson 2009). However, a higher lexical density can also be a sign of the opposite: that is, mirroring a less developed grammar and/or a language that is associated with younger children, demonstrating a "less sophisticated language" (Asker-Árnason et al. 2012).

Studies of lexical density have shown various results in regard to the DHH groups. Singleton (2004) included 72 DHH elementary students with low, moderate or high knowledge in American Sign Language (ASL). They were compared to hearing monolinguals of spoken English and hearing bilinguals of two spoken languages. All the DHH participants demonstrated a significantly higher lexical density (that is, fewer function words) as compared to their hearing comparison groups. However, the low-proficient signers showed a limited vocabulary and a high repetitiveness of some function words. The other proficient signers also struggled with their use of function words, but in comparison their texts were more creative, nonformulaic and diverse. It was suggested that high ASL proficiency may provide some learning advantages in English for the sign–print bilingual DHH group.

Another study of 18 Swedish DHH students (11–19 years old) using CI with spoken Swedish as the main language (the participants had limited knowledge in Swedish Sign Language) suggested the opposite. In this study, the DHH participants had a significantly higher lexical density in written narratives, compared to their hearing peers. The study suggests that since sign languages usually consist of many content words, this may have affected the written results for the students who had been exposed to sign language earlier in their lives (Asker-Árnason et al. 2012). Finally, a study comparing the writing by, on the one hand, DHH and, on the other hand, hearing adults showed no differences in lexical density between the groups (Wengelin 2002).

*Lexical diversity* is a measure that examines the lexical variation in a text and functions as an indicator of the writer's vocabulary. Both the above-mentioned studies by Singleton (2004) and by Asker-Árnason et al. (2012) also examined the lexical diversity in the written texts. The first study showed that the skilled signers of ASL compared equally to the monolingual participants regarding lexical diversity, while the texts by the low-proficient DHH signers had a lower lexical diversity, indicating that they used many of the same words repeatedly (Singleton 2004). The other study showed that the DHH participants using CI also had a significantly lower lexical diversity in their texts compared to their hearing peers (Asker-Árnason et al. 2012).

Finally, *spelling* is an aspect that often has been studied in written texts in studies including DHH and hearing children. Generally, spelling has been shown to be one of the DHH groups' strengths (Asker-Árnason et al. 2010) and this has been explained by the fact that they are more visually aware as they have more tools to use, such as fingerspelling in sign languages (Gärdenfors et al. 2019).

## 5. Writing Process

This section presents a review of earlier writing process studies including characteristic writing process features from a mono-, bilingual and an L2 view in both hearing, and DHH children and adults.

The field of cognitive writing processes has explored the three overarching, iterative processes: planning, translating and revision, since Hayes and Flower's influential model of writing in 1980. During the planning process, the writer sets a writing plan using information from her long-term memory and plans how to meet the goals in her writing. The translating process is the next stage in which she produces text. The last process, the revision process consists of reading, editing and evaluating the written text with the reason to improve it (Hayes and Flower 1980). As a child learns to write, she devotes much time to the low-level processes such as spelling, punctuation and grammar (translating process) by generating ideas directly into texts, resulting in limited resources to other more advanced writing tasks (McCutchen 1996). This kind of writing is even known as the knowledge-telling strategy (Kellogg 2008). However, as the low-level processes become automatized, more cognitive capacity is freed, giving the writers opportunities to advance

to next level(s), engaging in larger and more difficult tasks concurrent with translating, such as planning the content, revising or evaluating the text to the better. The main point of this is that limitations in the working memory capacity lead to the different focus in different developmental stages (McCutchen 1996, 2000; Kellogg 2008).

More fluent text production (as can be demonstrated by automatized typing) allows the writer to move beyond the knowledge-telling strategy and frees up more cognitive capacity for other tasks. However, this view may also lead to a paradox, since inexperienced writers often find it easy to generate text, while more experienced writers seem to set higher goals than just "text generating". These higher goals will mean using different information stored in long-term working memory, which would make the writing less fluent (McCutchen 2000).

A slower text production can be mirrored in more and longer pauses, that have often been the focus in studies of writing processes, with the underlying assumption that pauses will provide a "window" onto the cognitive processes underlying the language production (e.g., Flower and Hayes 1981; Wengelin 2006).

One study of writing processes showed that sign–print bilingual DHH adults (of STS and written Swedish) did not pause more often within words compared to hearing adults, and the kind of revisions they did was to correct typos (Wengelin 2002). Yet, another study reported that 11–19 years old monolingual DHH children of spoken Swedish had a slower writing flow (seconds per word) and showed a significantly higher percentage of pause time compared to their hearing controls. This was argued to reflect the fact that this group was more aware of their writing difficulties, resulting in a slower writing behavior (Asker-Árnason et al. 2010). Gärdenfors et al. (2019) investigated spelling in both the written product and the writing processes between groups of sign–print or bimodal bilingual deaf, hard of hearing, CI-users, CODAs and hearing monolinguals as well as unimodal bilinguals. The study showed that there were no differences in the spelling awareness (i.e., how likely writers are to detect and correct a spelling mistake).

Those few writing process studies on DHH groups are some exceptions, but the research body on writing processes in hearing children and adults is richer. A common finding among more experienced writers is that the more proficient a writer is, the fewer pauses will be found within words, as the writing flow will be faster (e.g., Schoonen et al. 2009; Palviainen et al. 2012).

Regarding revision, which is also a crucial part of the writing process: how much the writers revise the text by deleting or adding text may predict the quality of their final products. MacArthur (2016) summarizes: "revision is a critical aspect of writing that differentiates expert and novice writers" (p. 272) in which older and more proficient writers devote more time to revisions in order to improve their text, indicating a more aware writing behavior. Younger and less skilled writers generally revise less and make more revisions on the micro-levels (spelling, punctuations, etc.) rather than on macro-levels (sentences, paragraphs and improving the core of the texts).

Due to lack of accurate writing process studies from a bilingual perspective, some L2 studies will be accommodated, even if there may be differences between bilingual and L2 writing processes. Studies of L2 writing have reported that L2 writers show a similar behavior to younger writers. L2 writers carry out more revisions on micro-levels compared to their L1-peers, their writing fluency is more interrupted and the text quality is lower. This is often explained in terms of the L2 writers having an increased cognitive load because their basic writing skills such as spelling, grammar or typing are not automatized yet, which leaves them less time for the text content (Lindgren et al. 2008, 2019; Schoonen et al. 2009). However, the text quality does not always suffer even if the L2 writing requires a longer writing time and manifests a more interrupted text fluency and more revisions compared to the L1 writing; those writing processes will not always result in a lower text quality in the final texts (the written product). The L2 writers' need for more cognitive capacity could instead be compensated with a longer writing time, which may still result in well-performed final products (Lindgren et al. 2008).

The assumption behind this study is that a comparison of the written products between CODAs and the DHH group will result in more similarities than has been shown in earlier studies on children with hearing loss. This expectation is justified by the fact that the DHH and CODA groups in this study have similar language experience backgrounds. However, previous studies of writing processes have indicated that individuals with hearing loss as well as bilingual writers, respectively, show a higher cognitive load (expressed as more pausing) in their writing compared to hearing monolinguals. The other assumption is therefore that the DHH group will demonstrate a higher cognitive load in their writing. This is expected since the DHH group has hearing loss in addition to being bilingual.

## 6. Method

### 6.1. Participants

Twenty students between the ages of 10.7 and 12.8 (mean age: 11.6 years) were included in this study. All of them were bimodal bilinguals of spoken Swedish and STS and were recruited through hospitals, schools and networking. No one was reported to have further disabilities, such as reading or writing difficulties. They were divided into two groups: DHH children and CODAs. Each group consisted of ten students—seven girls and three boys, respectively. The great majority of the DHH and CODA children come from fluently signing families, explaining why the participants received high points on the STS test (see below), as the majority of the parents were DHH themselves. The children's signing knowledge was tested with SignRepL2, which is an established STS test (Schönström and Holmström 2017; Holmström 2018) applicable and used both for skilled and for non-skilled signers. On a point scale between 0 and 4, the DHH group had an average of 3.8 points, the CODAs had an average of 3.7 points. This is close to 4.0 points, which is the top score indicating a skilled signer. The independent sample *t*-test showed that there was no significant difference between the groups.

*DHH children.* The students (mean age: 11.7) were born with a hearing loss ranging between 40–110 dB without hearing aids or CI and 25–54 dB with their hearing technology, either CI or hearing aids. Six students used hearing aids and four students were bilaterally implanted (with the first operation ranging between 9 months and 2 years and 2 months). All were exposed to STS from 0 to 1.5 years of age and they grew up as bimodal bilinguals, mastering STS and spoken Swedish fluently. For nine DHH students, the hearing losses were discovered thanks to standard hearing screenings performed directly after birth. The tenth DHH student had her hearing loss discovered at the age of 1.2 years of age and started learning STS a few months later. Seven of them had DHH parents, two had skilled signing parents (who were educated STS interpreters), and one had parents with some signing knowledge. Six of the DHH children attended a special class for hard of hearing (HoH) where both spoken Swedish and STS were used to communicate, and the remaining three students were mainstreamed among hearing students. The students developed their spoken language through speech therapists, or through their hearing parents or relatives.

*CODA.* The students (mean age: 11.4) were hearing children of deaf parent(s). Seven CODAs had two DHH parents, and three had one DHH parent. All of them were exposed to STS from birth and they grew up as bimodal bilinguals, using spoken Swedish and STS daily. They have grown up in a signing environment where they have developed spoken Swedish simultaneously with help of hearing relatives, the other parent or in kindergarten. All CODAs were attending a school for hearing students with spoken Swedish as the language of instruction.

The parents filled out a questionnaire with background information about their children's school choice, hearing degree and language use. The complete metadata of the groups can be found in Table 1.

**Table 1.** The metadata of the participants. The table displays the participants' gender, age in years and months, SignRepL2 (STS-test) results (4.0 is the maximum), if they have DHH parent(s), their school choice (hearing class, or a hard of hearing, HoH-class), hearing technology (cochlear implants or hearing aids), if having CI–their implant age as well as their hearing degree with and without hearing technology.

| | Gender | Age | Sign-RepL2 | DHH Parent(s)? | Type of School | Hearing Technology | Implant Age | Hearing in dB with CI/HA | Hearing in dB without CI/HA |
|---|---|---|---|---|---|---|---|---|---|
| **DHH** | Boy | 10.7 | 3.88 | Yes | HoH-class | HA | | 40–54 | 55–69 |
| | Girl | 11.0 | 3.54 | No | Hearing class | CI | 1.2 years | 25–39 | >90 |
| | Girl | 11.1 | 3.86 | Yes | Hearing class | CI | 1.6 years | 25–39 | >90 |
| | Girl | 11.3 | 3.78 | No | HoH-class | CI | 0.9 years | 25–39 | >90 |
| | Girl | 11.4 | 3.80 | Yes | Hearing class | CI | 2.2 years | 25–39 | >90 |
| | Boy | 11.6 | 3.98 | Yes | HoH-class | HA | | n/a | 55–69 |
| | Girl | 12.0 | 3.84 | Yes | HoH-class | HA | | n/a | 55–69 |
| | Boy | 12.7 | 3.68 | No | HoH-class | HA | | 40–54 | 55–69 |
| | Girl | 12.8 | 3.98 | Yes | HoH-class | HA | | 25–39 | 40–54 |
| | Girl | 12.8 | 3.92 | Yes | Hearing class | HA | | 55–69 | 70–89 |
| **CODA** | Girl | 10.9 | 3.92 | Yes | Hearing class | | | | |
| | Boy | 11.0 | 3.78 | Yes | Hearing class | | | | |
| | Boy | 11.0 | 3.70 | Yes | Hearing class | | | | |
| | Boy | 11.0 | 3.92 | Yes | Hearing class | | | | |
| | Girl | 11.2 | 3.84 | Yes | Hearing class | | | | |
| | Girl | 11.3 | 3.62 | Yes | Hearing class | | | | |
| | Girl | 11.4 | 3.44 | Yes | Hearing class | | | | |
| | Girl | 11.6 | 3.76 | Yes | Hearing class | | | | |
| | Girl | 11.7 | 3.46 | Yes | Hearing class | | | | |
| | Girl | 12.5 | 3.52 | Yes | Hearing class | | | | |

### 6.2. Language Backgrounds

To ensure the participants' proficiency in STS and written Swedish, both languages were tested. In addition, the participants' parents and/or teachers filled in a background questionnaire. To summarize, all the participants showed an age-adequate language proficiency in both their languages, and we have every reason to consider them to be bilingual.

Related to this, some researchers have argued that it is only bimodal bilingual CODAs that can be "heritage learners" meaning that they learn a minority language at home (in this case, sign language) and they learn their dominant language (in this case, spoken language) through the community (see Polinsky 2018). The "heritage learners" concept cannot unproblematically be applied to the DHH children because their first language acquisition may be postponed, since most of them are born into non-signing families. Therefore, they will often show a delayed language development in their L1 as well as L2 (Mayberry 2007; Cormier et al. 2012). The DHH group in the present study has, however, received their hearing technologies and developed their L1 early, so their language situation is considerably closer to that of the CODAs. This is reinforced by the finding that this small group is similar to the CODAs regarding several literacy measures (Davidson et al. 2014; Goodwin et al. 2017; and Gärdenfors et al. 2019). This motivates why the DHH group also should be considered as heritage learners (see Chen-Pichler et al. 2018) and should be comparable to CODAs.

### 6.3. Writing Task and Procedure

The documents concerning the children's background information were handed over to the author by post or via the participants from their parents. The data collection mainly took place in a silent resource room at the schools in the presence of the author of this article. In some cases, data were collected at the students' homes in a silent room. The writing task was performed individually in a room where the students were shown a two-paged cartoon strip about Pink Panther (the cartoon strip can be found in the Supplementary Materials). They were instructed that they should use the pictures to create a written story,

and that the program they would use (that is, ScriptLog, see next section) had no grammar or spelling check. All students received the same instructions and were informed that they would perform the writing task on their own without any help or asking questions without any time limit. The procedure ended with the participants carrying out the SignRepL2 test (which took around 10 min).

On average, the writing task took 33.4 min for the DHH group and 32.1 min to complete for the CODA group (ranging between 12.81 and 58.25 min). The task was chosen in order to inspire and motivate the children to engage in the writing task. Previous studies (e.g., von Koss et al. 2016) provided 10 min for the participants (slightly younger than this age group) and found that this time limit restricted the possibility to assess the participants' narrative competence; the time pressure caused shortness and incompleteness of their texts. In addition, other studies of children's writing have successfully used a similar design, where the children were presented with a narrative task to solve without time pressure (Johansson 2009; Asker-Árnason et al. 2012).

### 6.4. Keylogging Tool

A keylogging tool, ScriptLog, was used to examine the writing processes, and the output from the program not only provides information about the students' writing process, but also their final products. To the writer, the program looks like a simple word processor. However, the tool logs everything the writer does on a computer including their text flow, writing time, total number of characters, pause patterns and revision behavior that may provide information about their strengths and struggles regarding their typing skills. Insights of the writers' writing process with help of their pause behavior provides a "window" to their cognitive processes underlying their language production (Wengelin 2006). Thanks to the automatic output of Scriptlog, the researcher is provided with very detailed statistical information about the writers' writing processes. The researcher can follow writing in real time as the program documents pausing behavior (such as their distribution of pauses before, within or after words and their pause percentage), writing speed and revisions, among other things (Leijten and Waes 2013).

### 7. Analysis

This analysis section covers writing outcomes measures of *the written product* and measures of *the writing process.* The information about the written product focuses on traditional properties while the information about the writing process focuses on online measures that can be generated from a keylogging tool. The measures are presented in detail in Tables 2 and 3.

### 7.1. Measures of the Written Product

Text measures come from the participants' final products and consists of four measures that are presented in Table 2: number of words, spelling errors, lexical diversity and lexical density.

### 7.2. Measures of the Writing Process

An essential part of the writing process is planning, which is often investigated through the examination of *pauses*, which can be obtained through a keylogging tool (ScriptLog in this case). However, it is not always easy to determine what a pause is, and when it occurs, so a researcher needs to adjust the pause criterion based on the study's research question. Both this study (see next paragraph for motivation), and many other studies sort out pauses shorter than two seconds (>2 s). If the criterion was set at 0 s, *any* pause, including millisecond short pauses would be included, resulting in data that would be overwhelming to analyze. Additionally, if a pause criterion is too low, false pauses may be included. To give an example, a young writer often needs more time to find a key compared to a more experienced writer, and if the researcher is not taking their slower writing time into consideration, those can accidentally be counted as "pauses", if the focus

is not to study the time it takes to find a key. To circumvent this, the transition time (the time it takes for a writer to find the next key) is often used as a guideline. The median transition time average (with a 0 s criterion) of the participants is 0.34 s (ranging between 0.17 and 0.73 s).

**Table 2.** The written product measures, with definitions.

| Measures | Definitions and Ways of Approach |
| --- | --- |
| **Number of Words** | The number of words based on the final products with help of Microsoft Word's word count. Words with incorrect spaces, for example *ytter dörren* instead of the correct *ytterdörren*, 'front door', were counted as one word. Names with spaces such as *Rosa pantern*, 'Pink Panther', were also counted as one word, so that they could be equated with personal names consisting of one word, such as *Rufus* or *Panter*. |
| **Spelling Errors** | Each misspelled word was counted as an error. In the next step the number of spelling errors was divided by the total of number of words. The measure is thus a proportion of spelling errors for all words. |
| **Lexical Diversity** | Lexical diversity reveals how varied a text is and works as an indicator of a writer's vocabulary: the more unique words a text has, the higher the lexical diversity. To count lexical diversity, the analysis program CLAN was used (MacWhinney 2000). Through this program, information about the lexical diversity through the ratio of unique words to the total number of words counted by means of VocD was generated (Malvern et al. 2004). |
| **Lexical Density** | Lexical density reveals how dense a text, if the text contains many lexical or grammatical words. The more lexical words, the higher the density, resulting in a more information-packed text (Halliday 1985; Johansson 2009). The content and function words were manually identified, and then extracted by means of CLAN's word count. The proportion of content words to the total number of words were then calculated in Excel (MacWhinney 2000). |

**Table 3.** The writing process measures, with definitions.

| Measures | Definitions and Ways of Approach |
| --- | --- |
| **Writing Time** | Information about the writing time between the pressing of the first and the last key. |
| **Characters in Total** | The total number of characters (including space, punctuation, comma, etc. that have been written during the whole writing session, including those removed). |
| **Deleted Characters** | A kind of revision. The number of deleted characters was divided by the number of the total number of characters in the text. |
| **Offline Text Flow** | Characters in final product divided by writing time in seconds. |
| **Online Text Flow** | Characters in total divided by writing time in seconds. |
| **Transition Time** | The average pause time between letters within a word—the time it takes for a writer to find a key. |
| **Pauses before Words** | This measure generates information about the time it on average takes to pause *before* writing a new word. The criterion was set to 2 s. |
| **Pauses within Words** | This measure generates information about the time it on average takes to pause within a word. The criterion was set to 2 s. |
| **Number of Pauses within Words** | This measure frames the number of pauses placed within a word that were longer than 2 s, without taking their length into consideration. |
| **Pause Percentage** | The amount of the writing session that consisted of pauses by dividing the total pause time with the total writing time. |

Pauses are here defined on an ad hoc basis, as 2 or more seconds of typing inactivity (Wengelin 2006), which gives a margin to all participants' transition times, since 2 s is almost three times as long as the slowest writer's transition time. The reason to choose median instead of mean is to exclude the longest pauses (such as when the participants look at the pictures and evaluating the text).

Based on this pause criterion, ten writing process measures with definitions are presented in Table 3: writing time, total number of characters, deleted characters, offline

and online text flow, transition time, pauses before and within word, number of pauses within words and pause percentage.

### 7.3. Statistical Analysis

Two statistical models were set in this study by using the statistical program R (R Core Team 2017). First, an independent *t*-test was set on all writing measures recently presented in order to compare the writing outcomes of the DHH and the CODA groups. Thereafter, a Pearson correlation analysis was set to examine the relations between the writing outcomes.

## 8. Results

Table 4 represents the results of each individual participant for all the measures including the measures of the writing process and the written product. Each groups' means are indicated in the table in gray. The last row indicates if there were any statistical difference between the groups. Since the result section showed few significant differences between the groups (only 2 effects out of 12 measures were significant), the reporting of significance level was extended from 0.05* to 0.1, in order to highlight the nuances of this statistical analysis. This led, however, to only one additional effect.

### 8.1. Group Level Statistical Analysis between DHH and CODA Children

Despite three quantitative differences found for *Lexical density, Deleted characters*, and *Spelling errors* between the DHH and CODA groups, the t-test revealed that only the first two reached significance.

Regarding *Lexical density*, the DHH children had a significantly higher lexical density (M = 0.44 in density, SD = 0.52) compared to the CODAs (M = 0.39 in density, SD = 0.34). The overall statistical model was set: t(2.531) = 15.478, *p* = 0.02*.

The second significant difference could be found in *deleted characters*. The two-sampled t-test showed that the DHH students deleted more characters (M = 18.6%, SD = 10.1%) compared to the CODAs (M = 11.3%, SD = 3.4%). The overall statistical model was set: t(11.06) = 2.18, *p* = 0.05*.

The last difference, which was not significant, could be found in the *Spelling errors*. The DHH children made fewer spelling errors (M = 1.9% spelling errors, SD = 1.5%) compared to the CODAs (M = 4.0% spelling errors, SD = 3.5%). The overall statistical model was set: t(12.312) = −1.784, *p* = 0.09.

### 8.2. Correlation Analysis

A Pearson correlation was performed on all the writing measures to show the relations (*R*-values) between all of the outcomes. The measures *pauses within words* and *spelling errors* showed no correlation so they were removed from this table. Observe that the correlations are presented vertically from the y-axes from Table 5.

Regarding *Signing knowledge* (a non-binary variable based on their SignRepL2 test results, see Table 1), three correlations were found: the more skilled the signers, the more deletions of characters (0.49*), the slower the offline text flow (−0.55*) and longer the pauses before words (−0.48*).

Regarding *Hearing* (as a binary variable, as DHH or hearing), two correlations were found. The hearing children (CODA) revised less (−0.46*) and demonstrated a lower *lexical density* (−0.51*) compared to their DHH peers.

Regarding *Characters in total*, five correlations were found: the more characters, the longer the writing time (0.69***), the shorter the *transition time* (−0.48*), the faster the online writing time (0.48*). The more characters, the greater the *number of words* (0.96***) and the higher the *lexical diversity* (0.67**).

**Table 4.** The writing outcomes including writing process and written product.

| Group | Characters in Total | Deleted Characters | Writing Time | Percentage Pausing | Transition Time | Offline Text Flow | Online Text Flow | Pauses before Words | Pauses within Words | Number of Pauses within Words | Number of Words | Spelling Errors | Lexical Diversity | Lexical Density |
|---|---|---|---|---|---|---|---|---|---|---|---|---|---|---|
| | | | | **Writing Process** | | | | | | | **Written Product** | | | |
| DHH | 2211 | 14.6% | 20.90 | 23.6% | 0.26 | 1.5 | 1.8 | 3.7 | 3.2 | 4 | 353 | 2.5% | 38.9 | 0.37 |
| DHH | 2500 | 28.1% | 33.01 | 41.3% | 0.25 | 0.9 | 1.3 | 2.9 | 2.3 | 5 | 318 | 4.4% | 54.1 | 0.43 |
| DHH | 2203 | 9.0% | 58.25 | 51.5% | 0.54 | 0.6 | 0.6 | 3.5 | 2.9 | 36 | 337 | 3.6% | 75.1 | 0.55 |
| DHH | 1413 | 14.3% | 31.70 | 48.2% | 0.58 | 0.6 | 0.7 | 4.1 | 2.4 | 19 | 243 | 2.5% | 39.2 | 0.46 |
| DHH | 1625 | 6.9% | 20.63 | 62.4% | 0.31 | 1.2 | 1.3 | 4.4 | 2.1 | 6 | 269 | 0.0% | 45.0 | 0.41 |
| DHH | 1803 | 14.0% | 31.23 | 43.9% | 0.35 | 0.8 | 1.0 | 3.4 | 3.6 | 4 | 297 | 3.0% | 59.3 | 0.44 |
| DHH | 2370 | 17.5% | 29.18 | 34.7% | 0.27 | 1.1 | 1.4 | 3.6 | 2.2 | 7 | 360 | 0.3% | 58.1 | 0.40 |
| DHH | 2197 | 23.7% | 39.14 | 46.3% | 0.36 | 0.7 | 0.9 | 3.9 | 2.9 | 15 | 322 | 0.9% | 52.5 | 0.42 |
| DHH | 1858 | 17.1% | 20.49 | 37.6% | 0.26 | 1.3 | 1.5 | 3.4 | 3.1 | 3 | 300 | 0.3% | 56.0 | 0.47 |
| DHH | 3718 | 41.1% | 49.66 | 48.1% | 0.17 | 0.7 | 1.2 | 3.1 | 4.5 | 8 | 394 | 1.0% | 71.0 | 0.48 |
| **Average** | **2189.8** | **18.6%** | **33.42** | **43.7%** | **0.33** | **0.9** | **1.2** | **3.6** | **2.9** | **10.7** | **319.3** | **1.9%** | **54.9** | **0.44** |
| CODA | 3067 | 9.0% | 44.13 | 54.7% | 0.25 | 1.1 | 1.2 | 5.1 | 2.4 | 10 | 522 | 0.6% | 56.5 | 0.37 |
| CODA | 1271 | 8.3% | 22.08 | 46.9% | 0.46 | 0.9 | 1.0 | 4.0 | 3.1 | 14 | 227 | 11.5% | 37.1 | 0.41 |
| CODA | 1638 | 11.2% | 21.87 | 50.0% | 0.27 | 1.1 | 1.2 | 3.8 | 4.6 | 3 | 290 | 4.1% | 38.3 | 0.39 |
| CODA | 2517 | 11.9% | 37.63 | 48.5% | 0.31 | 1.0 | 1.1 | 3.9 | 2.8 | 13 | 432 | 5.1% | 48.2 | 0.38 |
| CODA | 2178 | 17.1% | 25.40 | 41.6% | 0.24 | 1.2 | 1.4 | 3.6 | 2.2 | 3 | 345 | 0.9% | 54.3 | 0.41 |
| CODA | 2718 | 17.0% | 34.19 | 45.9% | 0.25 | 1.1 | 1.3 | 3.1 | 2.8 | 6 | 402 | 4.7% | 67.5 | 0.45 |
| CODA | 1284 | 6.8% | 16.92 | 39.5% | 0.31 | 1.2 | 1.3 | 3.1 | 2.7 | 10 | 225 | 1.8% | 32.0 | 0.37 |
| CODA | 2261 | 10.7% | 41.47 | 51.9% | 0.38 | 0.8 | 0.9 | 3.7 | 3.1 | 16 | 408 | 7.8% | 47.5 | 0.38 |
| CODA | 2083 | 9.5% | 19.64 | 45.2% | 0.19 | 1.6 | 1.8 | 4.1 | 2.6 | 4 | 335 | 0.3% | 51.0 | 0.43 |
| CODA | 6746 | 11.7% | 57.59 | 33.4% | 0.19 | 1.7 | 2.0 | 3.4 | 3.6 | 12 | 1155 | 3.5% | 74.0 | 0.34 |
| **Average** | **2576.3** | **11.3%** | **32.09** | **45.7%** | **0.28** | **1.2** | **1.3** | **3.8** | **3.0** | **9.1** | **434.1** | **4.0%** | **50.6** | **0.39** |
| **Significance** | | **0.05 \*** | | | | | | | | | | **0.09** | | **0.02 \*** |

\* Significance at $p \leq 0.05$ level.

**Table 5.** A correlation matrix of the examined measures. A * shows significance at $p \leq 0.05$ level, ** shows significance at $p \leq 0.01$ level, *** shows significance at $p \leq 0.001$ level. Dark gray indicates the correlations between the variables *signing knowledge* and *hearing* (DHH/CODA) and the writing outcomes. Light gray indicates the correlations between a written product and a writing process. The empty boxes show no correlations. OBS: The measures *pauses within words* and *spelling errors* were removed from the matrix because there were no correlations.

| | Signing Knowledge | Hearing | Characters in Total | Deleted Characters | Writing Time | Pause Percentage | Transition Time | Offline Text Flow | Online Text Flow | Pauses before Words | Number of Pauses within Words | Number of Words | Lexical Diversity | Lexical Density |
|---|---|---|---|---|---|---|---|---|---|---|---|---|---|---|
| Lexical Density | | −0.51 * | | | | | | | −0.49 * | | | | | |
| Lexical Diversity | | | 0.67 ** | | 0.76 *** | | | | | | | | 0.54 * | |
| Number of Words | | | 0.96 *** | | 0.62 ** | | | 0.46 * | 0.49 * | | | | | |
| Number of Pauses within Words | | | | | 0.61 ** | | 0.75 *** | −0.54 * | −0.64 ** | | | | | |
| Pauses before Words | −0.48 * | | | −0.48 * | | 0.47 * | | | | | | | | |
| Online Text Flow | | 0.48 * | | | | −0.60 ** | −0.80 *** | 0.94 *** | | | | | | |
| Offline Text Flow | −0.55 * | | | | | −0.49 * | −0.66 ** | | | | | | | |
| Transition Time | | | −0.48 * | | | | | | | | | | | |
| Pause Percentage | | | | | | | | | | | | | | |
| Writing Time | | | 0.69 *** | | | | | | | | | | | |
| Deleted Characters | 0.49 * | −0.46 * | | | | | | | | | | | | |
| Characters in Total | | | | | | | | | | | | | | |
| Hearing | | | | | | | | | | | | | | |
| Signing Knowledge | | | | | | | | | | | | | | |

Regarding *deleted characters*, a significant correlation was found with pauses before words: the more deleted characters, the shorter the pauses before words: (−0.48*).

Regarding *Writing time:* three significant correlations were found. The longer the *writing time*, the longer the *number pauses within words* (0.61**) and the greater the *number of words* (0.62**) and the higher the *lexical diversity* (0.76***).

Regarding the pause percentage, three significant correlations were found: the higher the pause percentage, the shorter the offline text flow (−0.49*), the shorter the online text flow (−0.60**) and the more pauses before words (0.47*).

Regarding *Transition time*, three correlations were found: the longer the *transition time*, the slower offline text flow (−0.66**), the slower the online text flow (−0.80***) and the more pauses within words (0.75***).

Regarding *Offline text flow*, three significant correlations were found: the faster the offline text flow, the faster the online text flow (0.94***), the fewer number of pauses within words (−0.54*) and the greater number of words (0.46*).

Regarding *Online text flow*, three significant correlations were found: the faster the offline text flow, the fewer the number of pauses within words (−0.65**), the greater the number of words (0.49*) and the lower the lexical density (−0.49*).

Regarding *number of words*, a correlation was found: the greater the number of words, the higher the lexical diversity (0.54*).

## 9. Discussion

The analyses of the written products in this study showed results in line with the findings from the few previous studies on DHH and CODA (Davidson et al. 2014; Goodwin et al. 2017; Gärdenfors et al. 2019); that is, that DHH and CODA who have a similar language experience background show few differences when the written product and writing process measures were compared. However, in contrast to the expectation that the DHH would use more pausing compared to the CODAs, this was not confirmed in this study. Nevertheless, the more skilled signers, regardless of group, had a slower writing flow and more pauses before words, which may be an indication of a higher cognitive load.

This study showed that both the DHH and CODA groups demonstrated many similar age-typical features in their texts with the exception of *lexical density* and *deleted characters*. The findings further indicate that it is important to differentiate between differences in linguistic background in the examined groups. With small numbers of participants in the study, it may be challenging to define patterns. In order to see possible tendencies, the *p*-value was increased to <0.1. This expanded *p*-value of the *t*-test revealed one more tendency, that the DHH group made fewer *spelling errors* compared to the CODA-group.

The first significant difference showed that the DHH children had a significantly higher *lexical density,* in which they used more content words compared to the CODAs. Increased lexical density has been shown by several earlier studies showing that DHH groups generally have higher lexical density compared to their hearing controls (e.g., Singleton 2004; Asker-Árnason et al. 2012). This contradicts earlier suggestions that a higher lexical density may be due to a transfer from sign languages that generally consist of more content words than function words (Asker-Árnason et al. 2010). In the present study, there was no difference regarding the signing knowledge of the DHH and the CODA group, but the latter group still had a lower lexical density. Thus, it is not necessarily the case that signing knowledge itself leads to higher lexical density in written Swedish. Instead, the differences in the groups' auditory input may be the reason. This argument is strengthened by the correlation analysis that showed a negative relationship between lexical density and the variable *hearing*, and not with signing knowledge. If we have as our starting point that a possible limited auditory input brings about a higher lexical density, the explanation may lie in the fact that spoken languages are more associated with a lower lexical density compared to written languages (Halliday 1985), reflected in the lexical density difference between the groups. Here, it is only the CODAs who have full access to the spoken language, giving them a greater opportunity to use, compare or transfer

patterns from the spoken to the written language because they do not have any hearing loss.

This suggestion is also strengthened by the fact that a high lexical density is mirrored in a slower online text flow. Producing a denser text (that is, a text that is more "written-like") is more likely to require more cognitive capacity as more time is needed to find the appropriate words, to evaluate, and revise the text. Content words also tend to be longer compared to function words as they are more frequent, which will likely require a longer processing time resulting in a slower lexical retrieval. This may therefore result in a slower text production (cf. McCutchen 2000) compared to a less dense text with short function words and faster lexical retrieval. This may thus explain why the children with higher lexical density have a slower text flow, because those texts are likely more cognitively demanding.

The second significant difference showed that the DHH children revised more (in the form of deleted characters) compared to the CODAs. In addition to the discussion above about the negative relationship between lexical density and text flow, similar findings have been interpreted as a more aware writing behavior (Asker-Árnason et al. 2010). The authors reported that their DHH participants (with spoken Swedish as primary language) performed significantly worse on almost all written product measures as they used significantly more pauses (pause percentage) compared to their hearing controls. The authors suggested that the greater use of pauses was a result of the DHH participants being more aware of their writing difficulties.

This study found no difference in pause percentage between the DHH and the CODA groups as there was almost no significant written product difference between the groups. Here, it may be the other way around: that the more deleted characters may lead to equal, or even in slightly better writing outcomes. Beyond the more deleted characters, the DHH group had slightly slower text flow measures (as shown by online and offline text flow) as they used more pauses within words. However, instead of showing difficulties in the written product, they showed equal, or even some better writing outcomes compared to the CODAs. The DHH group had a higher lexical diversity, and fewer spelling errors than their CODA peers. Lindgren et al. (2008) suggest that a slow writing behavior may not always result in a worse written product, but instead this may give the writers more time to reflect about their writing process such as word choice and how the words are spelled.

Regarding spelling errors, the statistical test indicated a tendency that is differentiated between the groups. The DHH children made more than half as many spelling errors compared to the CODA children. Paradoxically, they put more pauses within words compared to the CODAs. This would be interpreted as a more aware spelling behavior, strengthening the recent discussion about the fact that they put more time into the writing process (in the form of slower text flow) with the goal of improving the final product as much as possible.

Few spelling errors among this DHH group are in line with earlier studies (e.g., Gärdenfors et al. 2019) reporting that early, and proficient signing DHH students are often good at spelling. An explanation from this study is that their hearing loss, which usually results in difficulties in their phonological awareness (Ambrose et al. 2012), forces them to use other solutions than the sounding strategy in order to spell. As a result, this may have strengthened their orthographical memory instead as suggested by Gärdenfors et al. (2019). Another explanation proposed was that they are more likely to meet other signing adults (such as parents or teachers) who fingerspell to them, giving them a unique tool to circumvent the auditory strategy by establishing a link between fingerspelling and how a Swedish word is spelled (Gärdenfors et al. 2019).

Despite the recently discussed differences, the correlation analysis showed some overlapping information in which there also was a relationship between signing knowledge and some writing behavior: the better the signing knowledge (shown by higher points on the SignRepL2 test), the more deleted characters, the slower offline text flow and the longer pauses before words. As this correlation includes both DHH and CODA

children, the explanation here may lie in the fact that the most skilled signers have greater access to both languages compared to the "less" skilled signers, giving them greater repertoires to consider, compare or use information from one language (Williams and Lowrance-Faulhaber 2018). This may therefore have resulted in a higher degree of deleted characters/revision behavior, slower offline writing flow, longer pauses before words and a slower lexical retrieval (Lindgren et al. 2008; Schoonen et al. 2009). However, there is still a need to examine the revisions on a deeper level to see what is being revised (such as what is deleted, or added, and what text is moved between sentences/paragraphs). However, an interpretation of more revision is that the child processes the text more, which also indicates a more experienced writer.

Since the greatest number of correlations could be found between two written product measures or two writing process measures, and few between them, this indicates that there is likely no "norm" regarding how a particular writing process may lead to a particular written product. This study has, instead, shown that different writing processes may result in similar written products.

As the bimodal bilingual DHH and CODA groups of this study demonstrated few differences, in aligned with the previously presented studies (Davidson et al. 2014; Goodwin et al. 2017; and Gärdenfors et al. 2019), this shows the rationale of Hall and Dills' (2020) discussion. They suggested that many studies on DHH children have simplified, or not taken their varying language backgrounds into consideration, when comparing DHH with hearing children, in spite of the evidence that monolinguals and bilinguals have different psychological and linguistic starting points (Lindgren et al. 2008; Bialystok et al. 2008; Schoonen et al. 2009; Bialystok 2009; Duñabeitia et al. 2013; Giezen and Emmorey 2017).

The findings of this study indicate two things: first, that the impact of sign language on the writing outcomes seems to be applicable to all children, regardless of their hearing and language backgrounds. This leads to the second claim, that it is necessary to always have the participants' language backgrounds in consideration and include control groups with same language backgrounds as the examined group.

## 10. Conclusions

Before concluding with the study's most important findings, I want to highlight that although this group of participants is small, it is important to emphasize they come from a small population, and that the study consist of *very* many participants when considering the proportion of balanced bimodal bilingual DHH children in Sweden. They were also exceptionally well matched in terms of age and language background. The other similar but important American studies (Davidson et al. 2014; Goodwin et al. 2017) included between three and five bimodal bilingual DHH children, and the DHH population in the US is more than a hundred times as great as that in Sweden (Mitchell 2006). In this way, we have gained a unique knowledge about how bimodal bilingualism including sign language may affect literacy.

The study makes two important contributions. The first contribution is that unique bimodal bilingual insights in the DHH and CODA groups writing could be provided—not only in their written product but also in their writing process. This is probably the first time we could gain an insight into how a writing behavior results in a particular writing outcome among those bimodal bilingual groups with signing knowledge.

An important finding related to this contribution showed that both DHH and CODA groups with similar age and language backgrounds (including spoken Swedish and Swedish Sign Language) showed few differences on all measures, including the writing process and the written product. The few differences found were partly explained as a consequence of their different hearing backgrounds that may likely give them different opportunities to use the auditive input into the writing. Those few differences were most notable in the DHH group who, for instance, demonstrated a higher lexical density, deleted more characters and made fewer spelling errors compared to the CODAs. In addition, this study also demonstrates that written texts with similar profiles (number of words, lexical

properties, etc.) are not necessarily reflecting the same type of writing processes, and that the children use different processes to reach a similar final product. Yet, one result showed that signing knowledge was negatively correlated with offline text flow and pauses before words and positively correlated with deleted characters. This was explained in terms of the skilled signers having greater repertoires to consider, compare or use information from one language to the another, which may result in slower writing behavior due to more consideration while writing.

The second contribution of the study was to show the importance of taking the children's language background, in this case their bimodal bilingualism, into consideration. The study has shown that sign language knowledge correlates with some writing outcomes, so it can be deceptive to compare signing DHH children with hearing children with other language backgrounds, with the risk of pronounced differences. This should therefore always be guarded against through including true control groups with the same language backgrounds as the examined group.

Given that this study compared two bimodal bilingual groups, further research is needed in order to see what impact bimodality and bilingualism, respectively, have on writing, by including unimodal bilinguals, as well as including other groups such as hearing and DHH with a monolingual background.

**Supplementary Materials:** Cartoon strip is available online at https://www.mdpi.com/article/10.3 390/languages6020085/s1.

**Funding:** This research received no external funding.

**Institutional Review Board Statement:** The study was conducted according to the guidelines of Declaration of Helsinki, and approved by the Ethics Committee in Stockholm (2018/1033-31/5, 2018-06-19). Written informed consent to participate in this study and the questionnaires were provided by the participants' parents.

**Informed Consent Statement:** Informed consent was obtained from all subjects involved in the study. Written informed consent has been obtained from the subjects to publish this paper.

**Data Availability Statement:** The research data presented in this study are openly available in https://osf.io/4c7m3/ (accessed on 11 May 2021).

**Acknowledgments:** I want to express my greatest gratitude to my two supervisors, Krister Schönström and Victoria Johansson, for their continuous support during the study. I also want to thank the anonymous reviewers for valuable comments and feedback.

**Conflicts of Interest:** The author declares no conflict of interest.

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
