# Peer review of "The Writing Process and the Written Product in Bimodal Bilingual Deaf and Hard of Hearing Children"

_languages, doi:10.3390/languages6020085_

Round 1

Reviewer 1 Report

I think the paper in principle has a lot of potential but in my view,  sections outside of the study must be redone and one analysis should be added to increase strength of the argument.

1. In the first couple of the sections of the paper, the authors appear to suggest that sign-speech bilingualism and sign-print bilingualism can be treated on a par -- namely that it is the same type of phenomenon. I am not at all sure that this is the case and the issue deserves a discussion at the onset, which would simplify the discussion about the profiles. Summarizing: sing-speech bilinguals definitely be assumed to have proficiency in 2 languages in two modalities. The same cannot be said about sign-print bilinguals, for writing is not actually a natural language and represents one... but what exactly a writing system represents for a non-proficient learner without access to a spoken language is a different matter all together. In other words, this issue must be addressed right away and assumptions must be articulated.

Assuming then that DHH group in this study (somewhat non-traditionally) consists of individuals with access to sound vs individuals without such access (i.e. *deaf*), the authors are thus saying that unlike the *deaf* (who may very well be classified as the sign-print group only, whatever their knowledge of the actual natural language behind its written counterpart), the DHH and Codas are people with the access to the language itself which the written language is trying to represent. In this, at least in terms of examination of literacy development, we expect Codas and DHH to fall into one category -- to be comparable. 

This is sort of infused in various places in the paper but would be better to tackle from the beginning. This would also allow the readers unfamiliar with the complexities of the learner profiles to attain greater clarity.  [This is especially crucial given the authors' overall conclusions about (non-)comparability of profiles]

All of this is to say: I believe the intro, pre-study itself ought to be rewritten to reflect this issue directly, otherwise, it is not clear to what degree the references cited inform the debate. Additionally, without this direct re-angling, it appears that the authors miss a large amount of literature that is devoted to literacy instruction of the *deaf* (in their definition). However, if they simply say "look, we set aside kids without any access to hearing for the following reason, even though much of the lit also considers them bimodal bilingual", then their methodology is clear: these kids are actually uncontroversially exposed to the spoken language and its written representation a daily basis.

2. This brings me to the language analysis. It seems odd that a hanging fruit of bilingualism -- a language transfer kind of thing -- has not been mentioned. E.g, Wolbers '14 observe presence of ASL features in the English of deaf/DHH writers (Wolbers, K. A., Bowers, L. M., Dostal, H. M., & Graham, S. C. (2014). Deaf writers' application of American Sign Language knowledge to English. International Journal of Bilingual Education and Bilingualism17(4), 410-428.)

Given the reference to Goodwin 2017 as well as Davidson et al 2014, i would have thought something of this sort would have been said about the participants in this study. Have you seen any similarities in the the morphosyntax?

Here is the biggest part of the rationale:

a number of researchers have argued that Codas are bona fide Heritage learners (see the volume by Polinsky on these issues); it is not clear that we can say the same about DHH/CI population. One thing we know about Heritage learners is that their L1 patterns are different from other populations and in their dominant language they behave as native users(speakers/signers). If then, as is typically, Swedish Codas are Heritage language learners = native users of the dominant language, they should sound (if not look) native in their Swedish. The aforementioned is not true for the DHH/CI learners -- for them, the dominant language is unlikely to be a natural L1. It may depend whether the DHH child has a CI and has been signed to from birth (in which case this child has not been linguistically deprived and has had access to the powerful machine that has allowed analysis); if a child did not, then this is a delayed L1 acquisition (something like Kormier must be referenced). These issues need to be at least mentioned.   SO, if, in fact, there is a difference between Codas and DHH/CI in terms of morphosyntax, then the difference may be predictably attributable to the L1 status. 

Author Response

Dear Reviewer 1.

First of all, I want to say thank you for your time and your comments. I found them very helpful. I have colored each change in green. In the cover letter, you can find which changes I have made to meet your concerns. 

I hope that the point-by-point response (cover letter) will suit the manuscript. There may be a risk that the referred lines are different (some +/- lines) since I realized that they could be moving up-, or downwards when opening the manuscript from another computer. 

Please see the attachment to see the cover letter.

I am looking forward hearing from you. 

Best,
X

Reviewer 2 Report

The manuscript titled "The writing process and the written product in bimodal bilingual deaf and hard of hearing children" presents a unique study of how bimodal bilingualism - with a signed and spoken language - influences writing abilities by comparing hearing children of deaf parents and children who are deaf/hard of hearing. 

All aspects of the study are well-researched and well-presented, including the context, previous related research, research questions, methods and results. The findings are interpreted accurately and discussed from various perspectives in light of relevant literature.

I have no required revisions at this time.

Author Response

Dear Reviewer 2,

Thank you so much for your time reading the manuscript, and I am very happy to hear that you accept the manuscript in its current form. 

Wishing you the best.

Reviewer 3 Report

The present paper deals with the effect of bimodal bilingualism in writing process and written product. An experimental study is conducted on twenty deaf and hard of hearing (DHH) children and hearing children of deaf adults 6 (CODA) (mean 11;6 years) with the same bimodal bilingual backgrounds (a pint emphasized by the authors given a lack of control of this parameter in the previous literature). The experimental methodology (participants, procedure and results) is very well controlled and conducted. The results reveal quantitative but non significant differences between the two groups (except for the variable Lexical density) and some significant correlations. 

All my detailed questions are given throughout the annotated text (see the attached PDF document).

My general recommandations are 1/ to clearly present the main hypotheses relative to the writing processes and written product in bimodal bilinguals, the hypotheses relative to the two groups (what are the expected results given the previous ones), 2/ to be very prudent with the interpretation of the statistical effects (notably those which are not significant, be more precise with correlation analyses, signs are often missing) and 3/ to improve the discussion with regard to 1/ and 2/. 

Author Response

Dear Reviewer 3.

First of all, I want to say thank you for your time and your comments. I found them very helpful. I have colored each change in green. In the cover letter, you can find which changes I have made to meet your concerns. 

I hope that the point-by-point response (cover letter) will suit the manuscript. There may be a risk that the referred lines are different (some +/- lines) since I realized that they could be moving up-, or downwards when opening the manuscript from another computer.

Please see the attachment to see the cover letter.

I am looking forward to hearing from you. 

Best,
X

Round 2

Reviewer 3 Report

The authors answered to my questions and modified very correctly their paper. This study constitutes a very interesting contribution to the research field it addresses.

Author Response

Dear Reviewer,

This makes me happy to hear. Thank you so much for your feedback and comments, I found them very helpful.

Best wishes